# Training Spiking Neural Networks via Augmented Direct Feedback Alignment

**Yongbo Zhang**[1], **Katsuma Inoue**[1], **Mitsumasa Nakajima**[2],
**Toshikazu Hashimoto**[2], **Yasuo Kuniyoshi**[1], **and Kohei Nakajima**[1]
[1]Graduate School of Information Science and Technology, The University of Tokyo,
[2]NTT Device Technology Labs.
zhang@isi.imi.i.u-tokyo.ac.jp

## Abstract

Spiking neural networks (SNNs), the models inspired by the mechanisms of real neurons in the brain, transmit and represent information by employing discrete action potentials or spikes. The sparse, asynchronous properties of information processing make SNNs highly energy efficient, leading to SNNs being promising solutions for implementing neural networks in neuromorphic devices. However, the nondifferentiable nature of SNN neurons makes it a challenge to train them. The current training methods of SNNs that are based on error backpropagation (BP) and precisely designing surrogate gradient are difficult to implement and biologically implausible, hindering the implementation of SNNs on neuromorphic devices. Thus, it is important to train SNNs with a method that is both physically implementatable and biologically plausible. In this paper, we propose using augmented direct feedback alignment (aDFA), a gradient-free approach based on random projection, to train SNNs. This method requires only partial information of the forward process during training, so it is easy to implement and biologically plausible. We systematically demonstrate the feasibility of the proposed aDFA-SNNs scheme, propose its effective working range, and analyze its well-performing settings by employing genetic algorithm. We also analyze the impact of crucial features of SNNs on the scheme, thus demonstrating its superiority and stability over BP and conventional direct feedback alignment. Our scheme can achieve competitive performance without accurate prior knowledge about the utilized system, thus providing a valuable reference for physically training SNNs.

## 1 Introduction

Neuromorphic computing refers to a series of devices and models inspired by the real brain [1]. In the machine learning field, such biologically inspired technology is designed to simulate the learning and adaptability of the brain by utilizing hardware as accelerators to accomplish complex tasks with high accuracy and low energy consumption [1–4]. With the convergence of Moore's law and the increasing need for large-scale, low-energy neural networks, neuromorphic computing has great potential. Currently, although artificial neural networks (ANNs) have already achieved impressive performance on various tasks, the high computational complexity and energy consumption of ANNs hinder their application on neuromorphic devices [5]. Spiking neural networks (SNNs), which simulate the mechanism of real neurons in the brain, represent a solution to the application of neural networks in neuromorphic computing. Different from ANNs that use continuous scalars to represent and transfer information, SNNs communicate through streams of discrete action potentials or spikes, as shown in Fig.1a. This discrete spikes-based information processing mode makes SNN neurons consume energy only when they generate spikes, allowing SNN significantly reduce the activity times of neurons and energy demand for information transmission [6–8]. Taking advantage

38th Second Workshop on Machine Learning with New Compute Paradigms at NeurIPS 2024(MLNCP 2024).

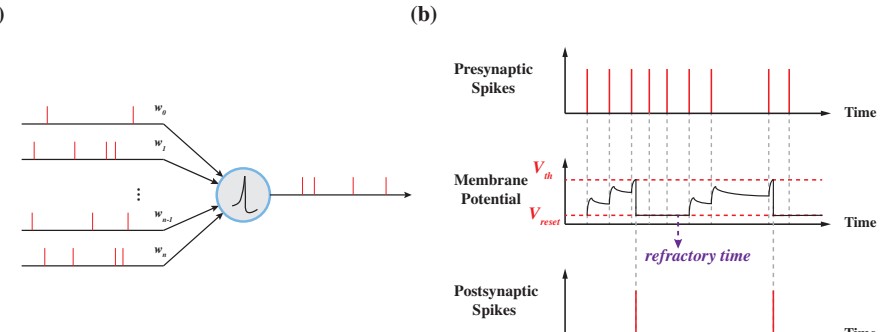

Figure 1: **The schematics of SNN neuron and dynamics of leaky integrate and fire (LIF) model.** **(a)** SNN neurons transmit discrete signals. **(b)** Presynaptic spikes are transmitted to postsynaptic neurons, leading to the accumulation of membrane potential, and the postsynaptic spikes are generated when the membrane potential exceeds the firing threshold. After this, the membrane potential is placed to reset the value, and SNN neurons enter a refractory period.

of low power consumption from simulated brain neurons, SNNs bridge the gap when it comes to the implementation of neural networks on neuromorphic devices.

As with ANNs, to obtain high-performance SNN models, effective optimization and training methods are essential. The most effective and representative training method in ANNs-gradient descent-based algorithm backpropagation (BP) [9]-has achieved remarkable success in many fields. However, because the dynamics of SNN neurons are described by discontinuous equations, their inability to perform gradient solutions makes BP difficult to be directly applied to SNNs, posing a challenge for training them. To address this training challenge, two main categories are proposed and widely adopted for porting BP to SNNs version: ANN-SNN conversion and surrogate gradient learning. ANN-SNN conversion methods convert the activation function of ANNs, which are pretrained by BP, into spiking activation mechanisms while keeping the trained parameters constant or using a weight balancing technique, which has achieved very high accuracy on many complicated tasks, such like image classification and speech recognition tasks [10–13]. However, because of the conversion mechanism, online learning and the physical implementation of these methods are not feasible. Surrogate gradient learning enables online learning of BP on SNNs by employing well-designed and accurately approximated differentiable functions to substitute the non-differentiable elements of SNN neurons during the backpropagation process. This training scheme uses flexible and efficient strategies to achieve excellent performance on several types of tasks [14, 15].

However, from the perspective of neuromorphic computing, BP-based methods are not the best choice. First, during the process of backward propagating errors, the networks need to fully record carefully orchestrated adjustments of all synaptic weights, leading to the physical implementation of these schemes being complex and unscalable [16–21]. Second, such a mechanism for transmitting all precise information layer by layer is considered biologically implausible [4, 22–25]. Considering the above problems, developing learning schemes for SNNs based on the neuromorphic computing idea is significant; that is, it is critical to develop easy-to-implement and biologically plausible algorithms.

Augmented direct feedback alignment (aDFA), a BP-free learning algorithm designed for physical neural networks, can be a promising candidate [26]. In aDFA, instead of transmitting error information layer by layer to update weights, as done in BP, the global error is injected directly into each layer through fixed, randomly initialized synaptic weights. Additionally, the arbitrary functions $g$ can be employed in the backward process rather than relying on $f'$, the exact derivative of the activation function. This approach, which breaks the BP transmission chain and uses imprecise information, is more implementable for physical platforms and is biologically plausible. From the aspect of SNNs, aDFA also can address the challenge posed by the nondifferentiable dynamics of SNN neurons. In [26], aDFA is preliminary applied to SNNs by employing $cos^2$ as the backward function. However, whether arbitrary functions can be used as backward functions for SNNs as in feedforward neural networks has not been systematically explored and confirmed, and the features of backward functions that can achieve better performance have not been analyzed. To use this

approach more efficiently and inform the training of SNNs implemented in neuromorphic devices, in the present study, we systematically validate the feasibility of the aDFA-SNNs scheme by using random functions with universal properties, present its range of validity, and analyze the settings of backward functions with genetic algorithm (GA) to achieve good performance. We also investigate the impact of basic but crucial features-network scale and temporal dynamics-of SNNs on our scheme, thus demonstrating the stability and superiority of it (see Appendix G and H). Finally, we directly adjust the parameter settings of the entire backward process without using forward information on the aDFA-SNNs schemes with certain forms of $g$, so that obtaining competitive performance (see Appendix I). Compared to existing studies, our scheme achieves a competitive performance while posing good hardware implementation feasibility (see Appendix K).

## 2  aDFA: BP, gradient-free training mechnism

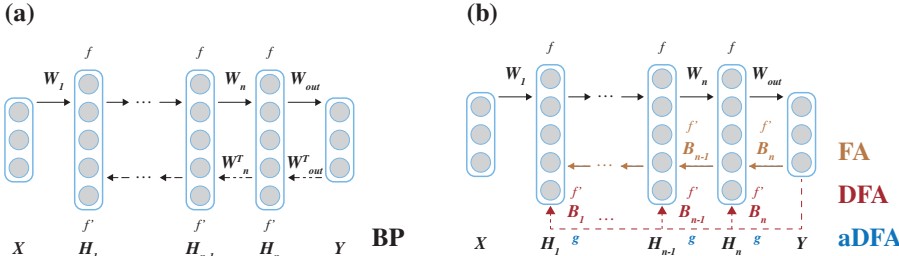

Figure 2: **Information flow of BP, DFA, and aDFA. (a)** BP, transmits the error signal layer by layer, needs to calculate precise $W^T$ and derivative $f'$. **(b)** Orange represents FA, where $W^T$ is replaced by fixed, randomly initialized matrices $B$. Red stands for DFA, which injects the global error from the last layer directly to each previous layer through fixed and random matrices $B$. Blue stands for aDFA, a drastic augmentation of DFA, substitutes for $f'$ by arbitrary nonlinear functions $g$.

Considering a standard multilayer network in Fig.2, the forward propagation is expressed as $x_{n+1} = f(a_n)$, where $a_n = W_n x_n$. $x_n \in \mathbf{R}^{N_n}$ is the input signal from $n-1$-th hidden layer $H_{n-1}$ to $n$-th layer, $N_n$ represents number of nodes in $H_n$. $W_n \in \mathbf{R}^{N_n \times N_{n-1}}$ stands for the weight for the $n$-th layer. $f$ denotes the element-wise activation function, which is often ReLu or sigmoid function in conventional ANNs [27–30], while in SNNs, $f$ is the non-continuous firing function [14, 31], as shown in Appendix A. In general, to train such a network, the connection matrices $W$ need to be optimized to minimize the loss function $L$. The process of the BP algorithm is shown in Fig.2a, here using the optimization of $W_n$ as an example, and the gradient $e_n$ that transmitted to $H_n$ through the chain-rule of BP can be expressed as:

$$e_n = \left[ W_{n+1}^T e_{n+1} \right] \odot f'(a_n), \tag{1}$$

where the superscript $T$ represents precise transposition, $\odot$ denotes Hadamard product, and $f'$ is the exact derivative of activation function $f$. With this information, we can compute the gradient of $W_n$ as $\delta W_n = -e_n \cdot x_n^T$. From Eq.1, we can see that the injected error signal to current layer depends on the error information from the layers behind, and it also needs to engage several precise calculations. Thus, this scheme is not the best choice from the perspective of neuromorphic computing. Feedback alignment (FA) [32], which is one of the earliest backward path-based BP-free algorithms, replaces the calculation of precise transposition in the backward process by employing fixed randomly generated matrices $B$, thus simplifying Eq.1 into:

$$e_n = [B_n e_{n+1}] \odot f'(a_n), \tag{2}$$

However, the sequential transmission mechanism still constrains the neuromorphic implementation of FA. Then, considering the solution of this mechanism, direct feedback alignment (DFA) [33–36], which can break the chain-rule of BP by injecting global error signal $e$ from the final layer to each previous layer directly through $B$, leads to a new mechanism:

$$e_n = [B_n e] \odot f'(a_n), \tag{3}$$

Nevertheless, the precise calculation of derivative $f'$ still could not be avoided, which impedes the complete physical implementation of this method. Additionally, in the context of its application on

SNNs, $f'$ cannot be obtained directly. Instead, it needs to be derived after accurately approximating the dynamics of SNN neurons into differentiable functions. Although several studies have successfully realized DFA-SNNs schemes [37–40], these accurate simulations and meticulous design processes are complex and require specialized expertise.

In aDFA, which is an impressive expansion of DFA, the $f'$ in Eq.3 is substituted with arbitrary nonlinear functions $g$, effectively addressing the derivative issue in DFA [26]. The training rule is then updated as:

$$e_n = [B_n e] \odot g(a_n). \tag{4}$$

Compared with Eq.1 in BP, Eq.4 can mitigate all terms, resulting in minimal feedforward information during training process. Breaking the BP chain-rule and the precise gradient calculation makes aDFA easy to implement physically and biologically plausible. Therefore, in the context of neuromorphic computing, aDFA is extremely suitable for training SNNs.

## 3 Results

First, we demonstrate the feasibility of the aDFA-SNNs scheme, which employs a variety of arbitrary nonlinear functions as backward functions $g$ in Eq.4, to check the effect of aDFA on the performance of SNNs. We use the benchmark task MNIST [41] with a simple three-layer fully connected SNN model. In this experiment, the model has dimensions $784 \times 1000 \times 10$, which consist of two spiking layers with 1000 and 10 nodes, respectively. The spiking layers are composed of leaky-integrate-and-fire (LIF) neurons (see Appendix A). For making a comparison, a smoother, exact approximation of the derivative of the discontinuous functions in LIF neuron-namely an approximation of the Dirac delta function-is utilized as the derivative $f'$ of the dynamics of LIF neurons during the backward process, thus constructing both standard BP and DFA schemes (see Appendix D).

For preparing nonlinear functions $g$, we generate them from random Fourier series (RFS) $g(a) = \sum_{k=1}^{k} [p_k sin(k\pi a) + q_k cos(k\pi a)]$, where $p_k$ and $q_k$ are random coefficients that are uniformly sampled from the interval $[-1, 1]$. $k$ is set to 4 in our case, and $p_k$ and $q_k$ are normalized by the relationship of $\sum_{k=1}^{k}(|p_k| + |q_k|) = 1$. As can be seen, RFS is the sum of a series of sine and cosine functions with introduced randomness, hence possessing the theoretical capability to indefinitely approximate any function. When generating RFS, we notice that neither the exact derivative of the dynamics of the LIF neuron, nor the smoother differentiable approximation $f'$ yields a negative value. Therefore, we introduce a shift to the vertical axis of RFS to obtain positive random Fourier series (PRFS). In fact, numerous standard RFSs are tested in this experiment; however, almost all of them proved to be ineffective. We consider that this phenomenon arises because the negative values of standard RFSs in the backward process change the updating direction of $W$, which affects the accumulation of membrane potential and firing of LIF neurons in the forward process, hence leading to training failure. We employ correlation coefficient $\eta$ (see Appendix E) to denote the degree of functional similarity between generated PRFS and $f'$ so as to conduct classified investigation and analysis on the performance of many generated PRFS on the aDFA-SNNs scheme. When $\eta$ equals 1, $g$ is the exact $f'$, that is, the standard BP and DFA cases; when it is 0, it represents uncorrelated case; and when it equals -1, it denotes the negative correlated case. In our study, the shape of $f'$, as indicated by the gray line in Fig.3c, is highly slender and distinctive, making it challenging to directly obtain $\eta$ with a higher value and broader range, which hinders a systematic classified analysis. Therefore, to achieve relatively higher value and wider range of degree of functional similarity with $f'$, we incorporate the scaling factor $\omega$ into PRFS to adjust its fundamental frequency. The transformed PRFS is presented as:

$$g(a) = |m| + \sum_{k=1}^{k} [p_k sin(\omega k\pi a) + q_k cos(\omega k\pi a)]. \tag{5}$$

where $|m|$ denotes a shift toward the positive field. To obtain proper $\omega$, that is, to achieve a higher value and wider range of $\eta$, we generate PRFSs with 10,000 random seeds, calculate their correlation coefficient $\eta$ with $f'$ at different orders of $\omega$, and investigate the distribution of them. The results are shown in Fig.3a. When $\omega$ equals 0.01, the distribution of $\eta$ is approximately normal in the range [-0.6, 0.6], which represents the maximum value and widest range of $\eta$ that we can obtain. We divide this range into six intervals with a uniform size of 0.2 and randomly select five PRFSs

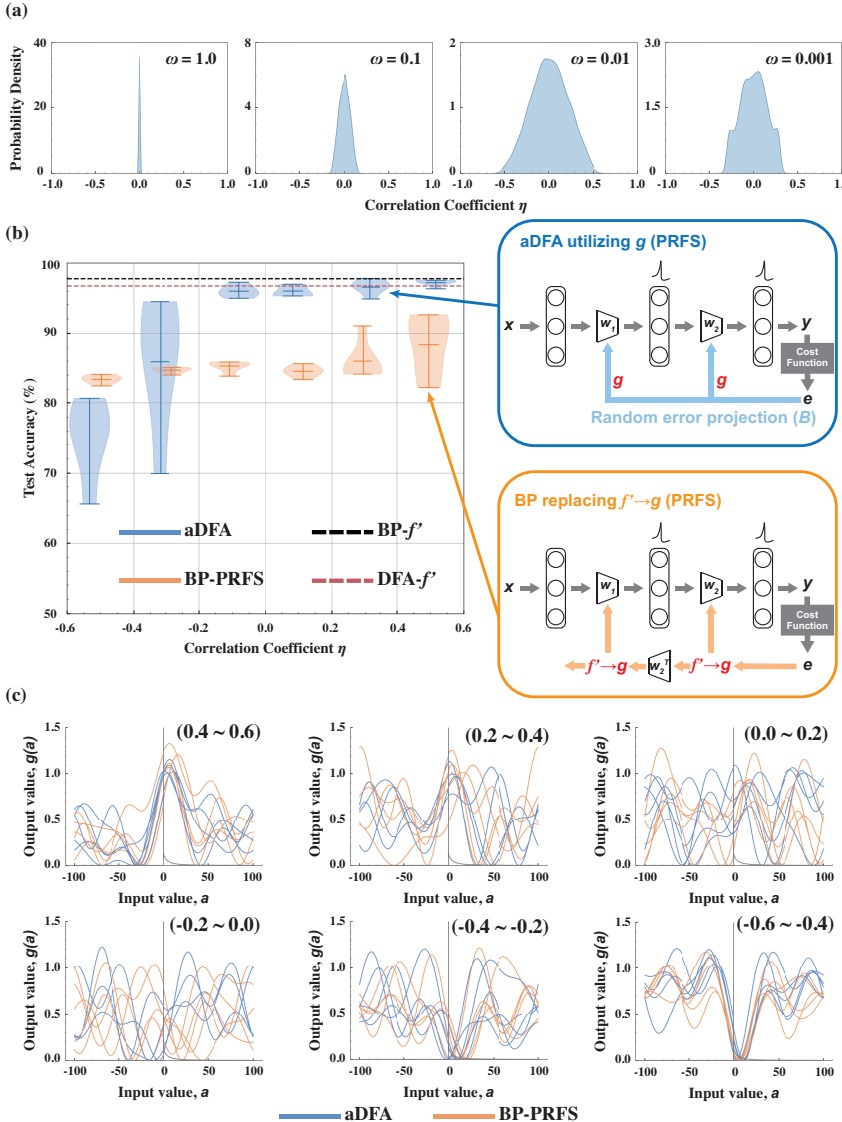

Figure 3: **Feasibility of the aDFA-SNNs scheme. (a)** The distribution of the correlation coefficient $\eta$ between PRFSs and $f'$ at different orders of scaling factor $\omega$. The $x$ axis represents values of $\eta$, and the $y$ axis denotes the probability density of distribution. **(b)** The test accuracy on MNIST task as a function of $\eta$ between $f'$ and PRFSs. The whiskers, the line in the middle of the box, and the filled area indicate the maximum and minimum values, the average value, and the distribution density, respectively. The dashed lines indicate the best performances of standard BP and DFA in five trials, which are 97.78% and 96.75%, respectively. **(c)** Figures of the corresponding shapes of PRFSs in each interval. The blue and orange lines represent selected PRFSs for aDFA and BP, respectively, and the gray line represents $f'$.

as $g$ within each interval to engage the aDFA-SNNs scheme. The experiment is also carried out on the BP-SNNs scheme for comparison, wherein randomly selected PRFSs are used instead of $f'$ during training. These schemes and the test accuracy as a function of $\eta$ are illustrated in Fig.3b. The whiskers, the line in the middle of the box, and the filled area indicate the maximum and minimum values, the average value, and the data distribution, respectively. The black and red dashed lines indicate the best performances of the standard BP and DFA on SNNs in five trials, at 97.78% and 96.75%, respectively (BP and DFA work unstably). The corresponding PRFSs that are randomly selected within each interval and $f'$ are illustrated in Fig.3c.

For BP with PRFSs, the average test accuracies are lower than 90%, regardless of the range of $\eta$, indicating the general failure of BP with randomly selected nonlinearities. On the other hand, when $\eta$ is greater than -0.2, aDFA can work stably and achieve good performance, even outperforming the best accuracy of standard BP, thus, demonstrating the effectiveness of aDFA on SNNs. The test accuracy of aDFA is significantly reduced and performs unstably when $g$ and $f'$ exhibit excessive negative correlation (i.e., $\eta < -0.2$). This is somewhat different from the conclusion of aDFA on feedforward neural networks in [26], where aDFA can work effectively by employing arbitrary $g$. We think this is caused by the fact that we shift $g$ to positive functions to avoid the effect of negative values on the accumulation and firing processes of LIF neurons. The results presented herein demonstrate the feasibility of the aDFA-SNNs framework, elucidate its effective working conditions, that is, by using positive nonlinear functions that are not excessively negatively correlated with $f'$ as backward function $g$, and show that the BP-SNNs scheme fails to utilize the mechanism of employing relaxed nonlinear functions. In general, to effectively train SNNs by using BP-based methods, it is necessary to approximate the dynamics of neurons so that we can carefully design the nonlinear functions in the backward process. While aDFA allows training of SNNs with relaxed nonlinear function, which do not contain any hyperparameter used in the dynamics of SNN neurons. This relaxed mechanism significantly alleviates the difficulty of physical implementation and allows avoiding the complex process of approximating SNN dynamics as the differentiable function.

We employ the genetic algorithm (GA), an evolutionary computational technique for updating and optimizing parameters [42–44], to search for parameter combinations of PRFS that achieve good performance in the aDFA-SNNs scheme described above. In this experiment, we randomly generate 10 PRFSs and use the test accuracy on MNIST and F-MNIST [45] tasks after one epoch of training as the fitness scores to optimize the random parameters $p_k$ and $q_k$ in Eq.5. The number of generations is set to 20. The performance of aDFA with GA-selected PRFSs, along with the standard BP and DFA schemes, are summarized in Table.1. As can be seen, the $f'$ based standard BP and DFA schemes exhibit unstable behavior and poor average performance on both tasks. Here aDFA scheme in our experiments demonstrates stable performance, and outperforms the standard BP and DFA schemes. Through GA, we also find that the initial irregular PRFSs always converge to shapes with a specific characteristic after evolution, that is, the "bell curve" near the peak of $f'$, as shown in Fig.A.1. Detailed information can be seen in Appendix F. Furthermore, information about the scheme's stability to changes in key features-network scale and temporal dynamics-of SNNs, and the competitive performance achieved by directly tuning backward process parameters, with comparisons to existing studies, is also presented in appendix.

| Framework | | MNIST | | F-MNIST | |
|---|---|---|---|---|---|
| Mechanism | Backward function | Best | Average | Best | Average |
| BP | $f'$ | 97.78% | 87.46% | 72.71% | 66.17% |
| DFA | $f'$ | 96.75% | 92.09% | 84.48% | 82.54% |
| aDFA | PRFS | **98.01%** | **97.91%** | **87.43%** | **87.20%** |

Table 1: **Performances of BP, DFA and aDFA.** Bold fonts indicate the best performances.

## 4 Conclusion

In the present study, we investigated the implementation of aDFA-a learning mechanism that is easy to implement physically and that is biologically plausible-on the SNNs platform. By using PRFS-random functions with universal properties-as the backward function $g$ to replace the meticulous designed derivative $f'$ of LIF neurons, we systematically showed the feasibility of the aDFA-SNNs scheme. We have presented the range of the validity of the approach and utilized GA to identify the PRFS settings that yield good performance. We also analyzed the impact of crucial features of SNNs on this scheme, so that showing the superiority and stability of it. Finally, we directly adjusted the $B$ and $g$ of schemes with determined forms of $g$, thus achieving competitive performance. Compared with BP and DFA, in our experiments, the stable and competitive performance obtained by the aDFA-SNNs scheme, which leverages the simple, straightforward, and hardware-friendly learning mechanism, provides a valuable reference for training SNNs. In the future, we will continue to explore the application of aDFA methods on more complex SNN models, and focus on developing general and efficient methods for optimizing backward functions $g$ to achieve competitive performance.

## Acknowledgements

We would like to acknowledge Shota Fujikawa and Ono Mikiya for their assistance in the early stage of the study. K. N. is supported by JSPS KAKENHI Grant Numbers 21KK0182 and 23K18472, by JST CREST Grant Number JPMJCR2014, and by Cross-ministerial Strategic Innovation Promotion Program (SIP) on "Integrated Health Care System" Grant Number JPJ012425.

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

# Appendix

## A Leaky-integrate-and-fire neuron

Numerous types of SNN neuron models have been proposed, but in the present study, we use one of the most popular mathematical neuron models called a leaky integrate and fire (LIF) neuron to construct our SNNs, which can achieve a good balance between the complexity needed to simulate dynamics of real neurons and the simplicity needed to model them [46, 47]. Fig.1b visually illustrates the dynamics of a single LIF neuron. For a given LIF neuron, the input-driving signal is derived from the weighted sum of the output of the spike sequences from all its connected presynapses, which is expressed as:

$$v(t)_i = \sum_j W_{ij} a(t)_j + b_i, \tag{A.1}$$

where $v(t)_i$ represents the input signal to a single neuron $i$ at time $t$. $a(t)_j$ is an output signal from presynaptic neuron $j$ at time $t$. $W_{ij}$ is the synaptic weight between neuron $i$ and neuron $j$, representing the strength of the connection. $b_i$ is injected bias.

The current membrane potential of the given LIF neuron, that is, the state of that neuron, depends on its previous membrane potential as well as the current input signal. To better numerically simulate this model, we consider its variation in discrete time, which leads to the dynamics of membrane potential being represented as:

$$h(t)_i = \left(1 - \frac{\Delta t}{\tau}\right) h(t-1)_i + \frac{\Delta t}{\tau} v(t)_i + \eta(t), \tag{A.2}$$

where $h(t)_i$ is the membrane potential of neuron $i$ at time $t$. $\Delta t$ represents the length of time step used in digital integration, and $\tau$ is the time constant used for the decay of membrane potential, both of which constitute the leaky factor. When $h(t)_i$ exceeds the threshold value, neuron $i$ will emit a spike to the postsynapses. The process of generating a spike output is expressed in the form of a piece-wise function as:

$$a(t)_i = \begin{cases} 1 & h(t)_i \geqq h_{th} \\ 0 & h(t)_i < h_{th} \end{cases}, \tag{A.3}$$

where $h_{th}$ is the threshold value of membrane potential. After neuron $i$ emits a spike, the membrane potential $h(t)_i$ of it will be placed to reset the value. The notation of $\eta(t)$ in Eq.A.2 is used to describe the reset process, which can be shown as:

$$\eta(t) = \begin{cases} 0 & h(t)_i < h_{th} \\ -\left\{\left(1 - \frac{\Delta t}{\tau}\right) h(t-1)_i + \frac{\Delta t}{\tau} v(t)_i\right\} & h(t)_i \geqq h_{th} \end{cases}, \tag{A.4}$$

where, when the current membrane potential $h(t)_i$ does not exceed threshold value $h_{th}$, the membrane potential will continuously accumulate, so $\eta(t)$ is placed at 0 so as not to affect the process of accumulation. When $h(t)_i$ exceeds $h_{th}$, for simplifying the simulation and making the model more generalized, we set the reset value to 0, meaning $\eta(t)$ is placed at the negative of the current membrane potential $h(t)_i$ of neuron $i$. After the reset process, the neuron will enter the refractory period, during which $h(t)_i$ does not follow Eq.A.2 but remains being pinned at 0, preventing the neuron from being fired.

## B Experimental Setup

In our study, we employ an identical LIF model and initialization techniques of weight matrices $W$ and backward matrices $B$ as in a previous broadcast alignment (BA) paper[37]. BA is a variant of DFA, which has been utilized to achieve good performance on SNNs. We utilize the benchmark datasets MNIST and Fashion-MNIST for image classification to assess the performance of the proposed framework [41, 45]. The inputs are not encoded; instead, a direct mapping method is employed to continuously inject static input signals in a period of interval so as to fulfill the requirements of time dynamics in LIF neurons and the simplicity and universality of experiments. The duration of the time interval $T$ is set to 100 ms and divided into two segments. The first segment is a 20 ms running period, during which we keep injecting input signals to obtain stable states of the

LIF neurons. The second segment is the 80 ms training period, in which we start training connection matrices $W$ to optimize the performance of the network. The output is defined as the one-hot label that corresponds to the most active neuron in the output layer, that is, those that generate the highest number of spikes during both the training and testing phases. The error information $e$ is determined by calculating the difference between the target outputs and predictions of the network, hence ensuring adherence to the principle of standard DFA. Each model in our study is trained for 20 epochs, and the size of the minibatch is set to 100. The learning rate is specific to each layer, and the learning rate of each layer is inversely proportional to its input dimension.

For modeling the LIF neuron, the threshold value of the membrane potential $h_{th}$ is set to 0.4 while the length of time step $\Delta t$, refractory time $t_{ref}$, and time constant $\tau$ are set to 0.25 ms, 1 ms and 20 ms, respectively. The initialization of fixed random matrices $B$ is performed using the following method:

$$B_n = \gamma^D \prod_{i=n}^{n+D} \left[ \bar{W}_{n+1} + 2\sqrt{3}\sigma_{W_{n+1}} \left( rand - 0.5 \right) \right] \tag{A.5}$$

where $B_n$ represents the fixed random mapping to layer $H_n$, $\bar{W}_{n+1}$ is the desired mean of connection weights, $\sigma_{W_{n+1}}$ denotes the standard deviation of $W_{n+1}$ (the initialization method of $W$ is included in the Appendix C), and $rand$ has a uniform distribution over the range [0,1]. The variable $D$ represents the number of downstream layers, while $\gamma$ is the scale factor that adjusts the range of values in fixed random mapping. Specifically, $\gamma$ is set to a constant value of 0.0338.

## C  Initialization of matrices

In order to make comparison with the existing biologically plausible SNN training framework, the broadcast alignment (BA), we use the same initialization method for fixed random matrices $B$ and connection matrices $W$ [37]. This initialization method is similar to the techniques in computer science, rather than them in real brain. The biases $b$ are initialized to a physiological value of 0.8. The weight matrix in the $n$-th layer is initialized as follows:

$$W_n = \bar{W}_n + 2\sqrt{3}\sigma_{W_n}(rand - 0.5), \tag{A.6}$$

where, $rand$ has a uniform distribution over the range [0, 1], $\sigma_{W_n}$ represents the standard deviation of the weights $W_n$, $\bar{W}_n$ denotes the desired mean of the weights, and the desired second moment of weights $\bar{\bar{W}}_n$ are expressed as:

$$\bar{W}_n = \frac{(\bar{v} - 0.8)}{(\alpha N \bar{v})}, \tag{A.7}$$

$$\bar{\bar{W}}_n = \frac{(\bar{\bar{v}} + \alpha^2 (N - N^2)\bar{W}_n^2 \bar{v}^2 - 1.6\alpha N \bar{v}\bar{W}_n - 0.64)}{(\alpha^2 N \bar{\bar{v}})}, \tag{A.8}$$

where, $\bar{v}$ and $\bar{\bar{v}}$ denote the mean value and second moment of value of input signals, respectively, with values of 8, 164. $N$ represents the number of nodes in the $n$-th layer, $\alpha$ is constant with value of 0.066. $\sigma_{W_n}$ in Eq.A.6 can be calculated by $\bar{W}_n$ and $\bar{\bar{W}}_n$. It should be noted that the $\bar{W}_n$ and $\sigma_{W_n}$ also will be employed to initialize $B$. The values of $W$ and $B$ initialized in this way will be within a reasonable range i.e. not too large and not too small for the SNNs.

## D  Accurate differentiable approximation of LIF neurons

For making a comparison, a smoother, more exact approximation of the derivative of the discontinuous functions in the LIF neuron, that is, the approximation of the Dirac delta function, is utilized as the derivative $f'$ of the activation function during the backward process, thus constructing both standard BP and DFA frameworks. The $f'$ is expressed as:

$$f'(a) = \begin{cases} \frac{h_{th} t_{ref} \tau}{a(a - h_{th})\left(t_{ref} + \tau \log\left(\frac{a}{a - h_{th}}\right)\right)^2} & a > h_{th} \\ 0 & a \leqq h_{th} \end{cases}, \tag{A.9}$$

where the input to the function is represented by $a$, while the values of $h_{th}$, $t_{ref}$, and $\tau$ are given in Section B. The above experiments are also conducted on the standard BP and DFA frameworks

that have been constructed in this manner for comparative analysis; the results demonstrate the effectiveness, stability, and superiority of our aDFA-SNNs framework.

# E   Correlation coefficient

We employ the correlation coefficient $\eta$ to denote the degree of functional similarity between the generated PRFS and $f'$ so as to conduct a classified investigation and analysis on the performance of numerous generated PRFS on the aDFA-SNNs framework. The expression of $\eta$ is shown as:

$$\eta = \frac{\int \left\{ f'(a) - \overline{f'}(a) \right\} \left\{ g(a) - \overline{g}(a) \right\} da}{\sqrt{\int \left| f'(a) - \overline{f'}(a) \right|^2 da} \sqrt{\int \left| g(a) - \overline{g}(a) \right|^2 da}}, \tag{A.10}$$

where $g\left( a \right)$ represents generated PRFS, the superscript mean the average, and the range of integration is set as $[-100, 100]$. When $\eta$ equals 1, $g$ is the same as $f'$, that is, the standard BP and DFA cases; when it is 0, it represents the uncorrelated case; and when it equals -1, it denotes the negative correlated case.

# F   Using genetic algorithm to obtain well-performing settings

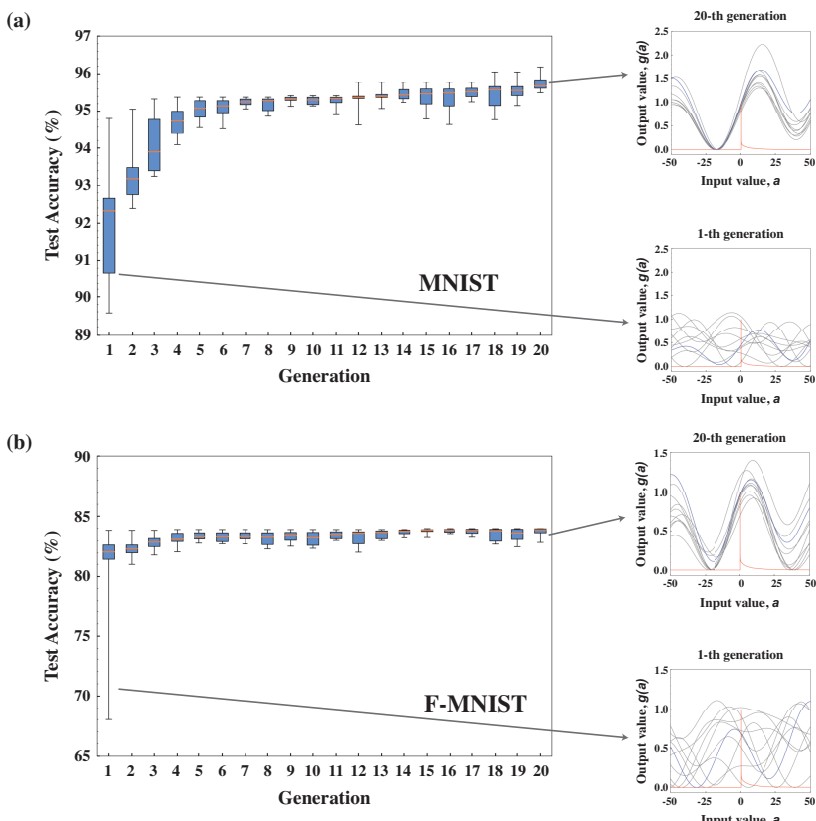

Figure A.1: **The results of the genetic algorithm (GA) optimizing and evolving PRFSs.** The left figures denote the fitness score as a function of generation, showing the evolutionary processes of PRFSs. The fitness score is represented by the test accuracy. The box, whisker, and orange line represent the distribution, maximum and minimum values, and median of the population's fitness score, respectively. The right figures represent the shape of PRFSs for randomly initialized and final generation. The red line represents the smoother approximation derivative $f'$, the gray line represents the PRFSs in the population, and the blue line represents the best-performing individual PRFSs. **(a)** Results on MNIST. **(b)** Results on F-MNIST.

In this section, we investigate the general feature of backward nonlinear functions $g$ that can yield good performance in the aDFA-SNNs scheme. We employ PRFSs, as illustrated in Eq.5, as the backward nonlinear functions $g$ of the fully connected aDFA-SNNs framework with a dimension of $784 \times 1000 \times 10$. The genetic algorithm (GA), which is a evolutionary computational technique for updating and optimizing parameters [42–44], is subsequently utilized to search for good PRFS parameter combinations to acquire appropriate nonlinear functions that can achieve good performance.

In this experiment, we randomly generate 10 PRFSs, that is, the population is set to 10, and use the test accuracy on the MNIST and F-MNIST datasets after one epoch of training as the fitness scores to optimize the random parameters $p_k$ and $q_k$ in PRFSs. The number of generations is set to 20, and in each generation, the two highest-scoring individuals will undergo crossover and mutation processes to generate offspring that replace the worst-performing individual in the population. The evolutionary processes of PRFSs, that is, the results of the population's fitness score as the function of generation, are depicted in the Fig.A.1. The box, whisker, and orange line represent the distribution, maximum and minimum values, and median of the population's fitness score, respectively. The shape of PRFSs for randomly initialized and final generation are plotted in the left figures of Fig.A.1. The red line represents the smoother approximation derivative $f'$, the gray line represents the PRFSs in the population, and the blue line represents the best-performing individual of PRFSs. As can be seen, as the number of generations rises, which indicates the evolution process, the fitness scores improve while the data dispersion decreases on both tasks; this means that the performance of the aDFA-SNNs scheme with PRFS becomes better and more stable. This observation shows the successful evolution of PRFS. Therefore, by utilizing this method of automatically updating and evolving parameters, we can obtain proper settings for PRFS that can achieve good performance. From the PRFS shapes, the initial irregular PRFSs always converge to shapes with a specific characteristic after 20 generations, that is, the "bell curve" near the peak of $f'$. The average test accuracy of the best-scoring individuals in the final generation through 20 epochs of training can reach 97.91% and 87.20% on the MNIST and F-MNIST datasets, respectively (the selected GA-PRFSs are used to conduct five trials). These results suggest the general feature of PRFS that can achieve good performance in the aDFA-SNNs scheme is possessing a "bell curve" shape when their input values are near the threshold value of membrane potential of the SNN neurons.

## G  Impact of the network scale

In this section, we investigate the impact of network size, one of the most fundamental and crucial characteristics of neural networks, on the aDFA-SNNs scheme. We employ a three-layer fully connected SNN model with the similar architecture, and same experimental settings as that in the previous experiments, and utilize the number of nodes within the hidden layer to denote the size of the network. We also examine the impact of this characteristic on standard BP and DFA methodologies for making comparison. The frameworks are evaluated by using both MNIST and F-MNIST datasets. For the aDFA-SNNs scheme, we conduct experiments with five randomly selected PRFSs in the range of $\eta$ belonging to [0.4, 0.6], while for the standard BP and DFA frameworks, five trials are conducted with $f'$. The results of testing accuracy as a function of the number of nodes in hidden layer are shown in Fig.A.2. The box plots show the data distribution of frameworks to illustrate their stability. Whiskers, orange lines, box bodies, and dots represent the maximum and minimum values, median, data distribution, and outliers, respectively. The line chart illustrates the mean test accuracy of examined frameworks, serving as the indicator to reflect their performance and trends, while facilitating a direct comparison. These results demonstrate that the aDFA-SNNs scheme can work stably and achieve good performance on both datasets, regardless of network size. Furthermore, as the network size increases, there is a consistent improvement in test accuracy leading to eventual convergence. The standard BP and DFA frameworks, however, exhibit significant instability and poor performance on both datasets, failing to show the dependency of test accuracy on network size. In addition, the average performances of aDFA surpass that of standard DFA on both datasets, irrespective of the network size. Only in small network size, standard BP can achieve competitive or better test accuracy than aDFA. Therefore, the analysis and comparison of this characteristic demonstrate the exceptional stability of the aDFA-SNNs scheme, and show that aDFA is more suitable for large-scale SNNs than BP and DFA, which highlights the superiority of aDFA-SNNs scheme.

**(a) MNIST**

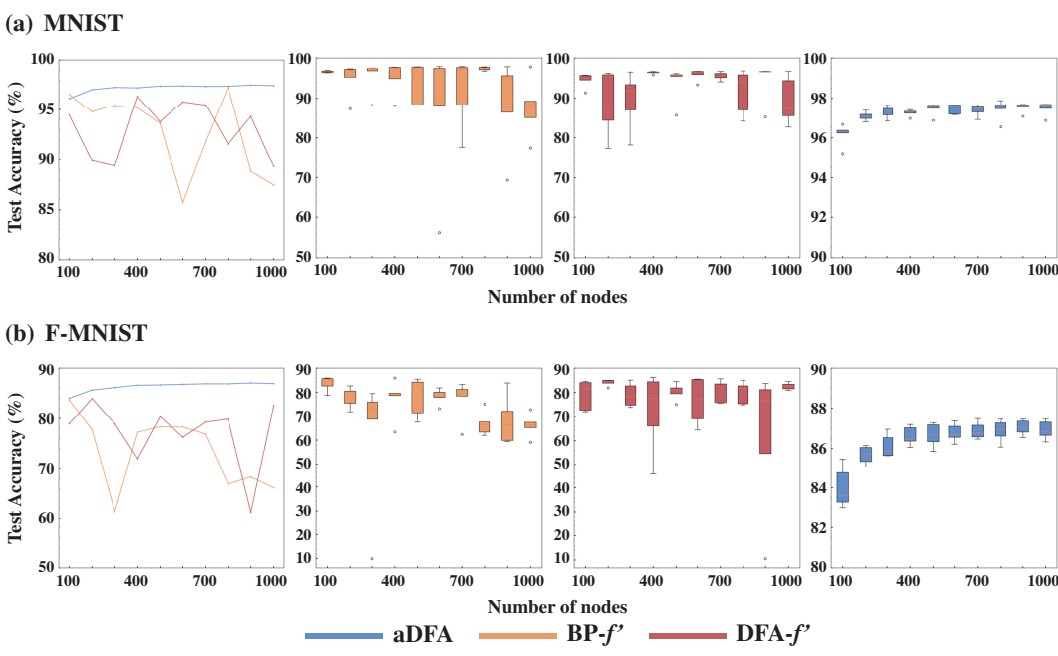

**(b) F-MNIST**

aDFA     BP-*f'*     DFA-*f'*

Figure A.2: **The results of the impact of network size on the performances of aDFA-SNNs scheme.** The size of the network is represented by number of nodes in hidden layer. The box plots show the data distribution of frameworks. Whiskers, orange lines, box bodies, and dots represent the maximum and minimum values, median, data distribution, and outliers, respectively. The line chart illustrates the mean test accuracy of examined frameworks. **(a)** The results on the MNIST task. **(b)** The results on the F-MNIST task.

# H   Impact of the temporal dynamics

Another characteristic being analyzed is the temporal dynamics of LIF neurons. In this experiment, we alter the temporal dynamics of LIF neurons by changing their length of time step $\Delta t$. By measuring the test accuracy of three-layer fully connected SNN model, which with dimension $784 \times 1000 \times 10$, on MNIST and F-MNIST datasets as $\Delta t$ is varied, we investigate the robustness of the aDFA-SNNs scheme to the impact that from changing the temporal dynamics of LIF neurons. We employ the identical approach as in sectionG to conduct experiments, wherein we utilize the same PRFSs on the aDFA-SNNs scheme, and also employ the standard BP and DFA frameworks with five 5 trials for making comparison. The results are shown in Fig.A.3. The box plots illustrate the data distribution, while the line plots depict the average test accuracy of frameworks as a function of $\Delta t$. Here, the refractory time $t_{ref}$ of 1ms is a critical factor that requires our attention. Specifically, when the length of time step $\Delta t$ exceeds or equals 1ms, LIF neurons will lose their refractory period, resulting in significant alterations in their temporal dynamics. The phenomenon above is evident in our findings, where the test accuracy of all investigated frameworks exhibit significant decreases when $\Delta t \geqq 1$ms. For standard BP-SNNs and DFA-SNNs frameworks, when the temporal dynamics are significantly altered, they are failing to achieve meaningful learning. On the other hand, although the aDFA-SNNs scheme experiences a drastic reduction in test accuracy, it can still exhibit learning capabilities to a certain degree, with average accuracy of more than 90% on MNIST. When the $\Delta t$ less than 1ms, the aDFA-SNNs framework demonstrates stable performances and achieves high accuracy, with mean accuracy of approximately 97% on MNIST dataset. In contrast, the dispersion of the test accuracy distribution observed in standard BP-SNNs and DFA-SNNs frameworks indicate unstable performances, and their mean test accuracy presented in the line graphs indicate mediocre performances. In addition, these results also demonstrate the high sensitivity of the BP-SNNs framework to changes in temporal dynamics of LIF neurons, that is, the variations of $\Delta t$ have greater impacts on the performances of it. For DFA-SNNs and aDFA-SNNs frameworks that based on the mechanism of direct error transmission with random mappings, they are robust to non-significant changes in this characteristic. Specifically, when $\Delta t <$1ms, the

**(a) MNIST**

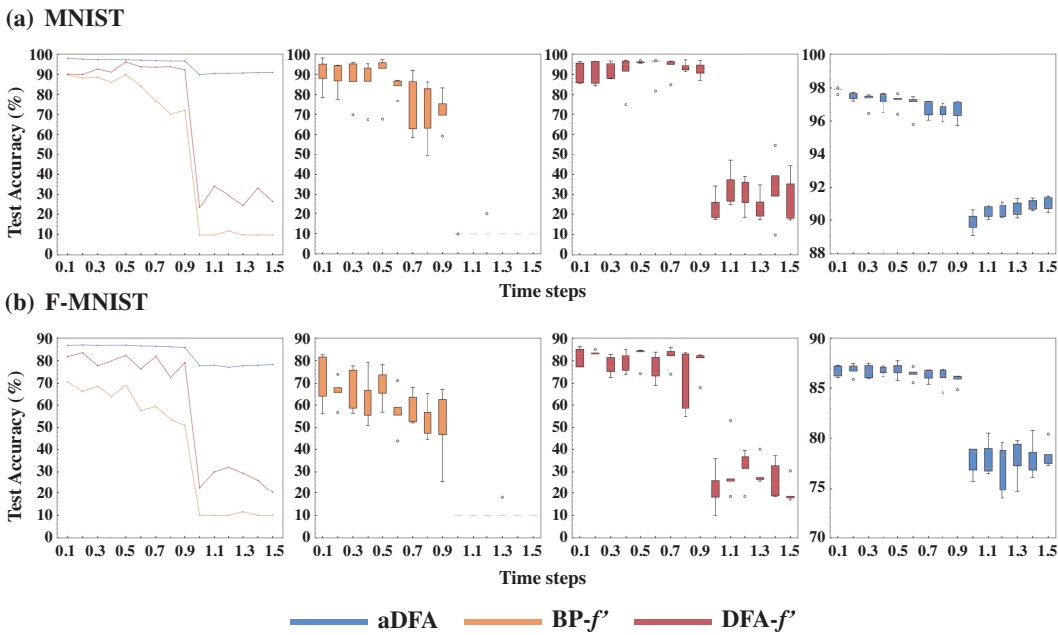

**(b) F-MNIST**

Figure A.3: **The results of the impact of temporal dynamics of the LIF neuron.** The time steps $\Delta t$ is used to represent the changing of temporal dynamics in LIF neurons. The box plots show the data distribution of frameworks. Whiskers, orange lines, box bodies, and dots represent the maximum and minimum values, median, data distribution, and outliers, respectively. The line chart illustrates the mean test accuracy of examined frameworks. **(a)** The results on the MNIST task. **(b)** The results on the F-MNIST task.

changes of $\Delta t$ have little influences on their performances. In general, BP-SNNs is highly sensitive to the characteristic of temporal dynamics, while DFA-SNNs exhibits robustness towards to non-significant changes of it. The aDFA-SNNs framework can not only maintain stability and achieve good performances under non-significant variations of temporal dynamics, but also exhibit a certain degree of robustness to drastic changes of it. This superiority demonstrate the flexibility and simplicity of the aDFA-SNNs framework in designing parameters of LIF neurons, as well as the applicability and reliability of its physical implementation. Therefore, it is further elucidated that aDFA-SNNs scheme aligns with the principle of Neuromorphic Computing.

# I Explorations of the performance of backward functions with fixed nonlinear form

The error transmission in the backward process of aDFA method involves two crucial relaxed components, namely the fixed random mappings and the arbitrary backward nonlinear functions, denoted as $B$ and $g$ in Eq.4 respectively. Unlike in standard BP method, which employs strictly exact $W^T$ and sequential transmission as the error mapping mechanism, and unlike in both BP and DFA methods, which take the precise derivative of the activation function as the backward nonlinearity. The utilization of a relaxed error transmission mechanism by aDFA provides an invaluable opportunity to directly adjust the entire backward process, thereby further enhancing the performance of networks. In other words, by using the aDFA method, the $B$ and $g$ in Eq.4 can be directly adjusted to improve the performance of the network, regardless of any information in the feedforward process. Based on this principle, in this section, we employ two nonlinear functions with determined form as the backward function $g$ to construct aDFA-SNNs schemes, then directly adjust their scale factors $\gamma$ in the initialization of $B$ (shown in the Eq.A.5) as well as parameters of selected nonlinear functions $g$, leading to achieve competitive performances with high neuromorphic hardware feasibility. The first function is the Gaussian function, commonly employed in surrogate gradient learning as an

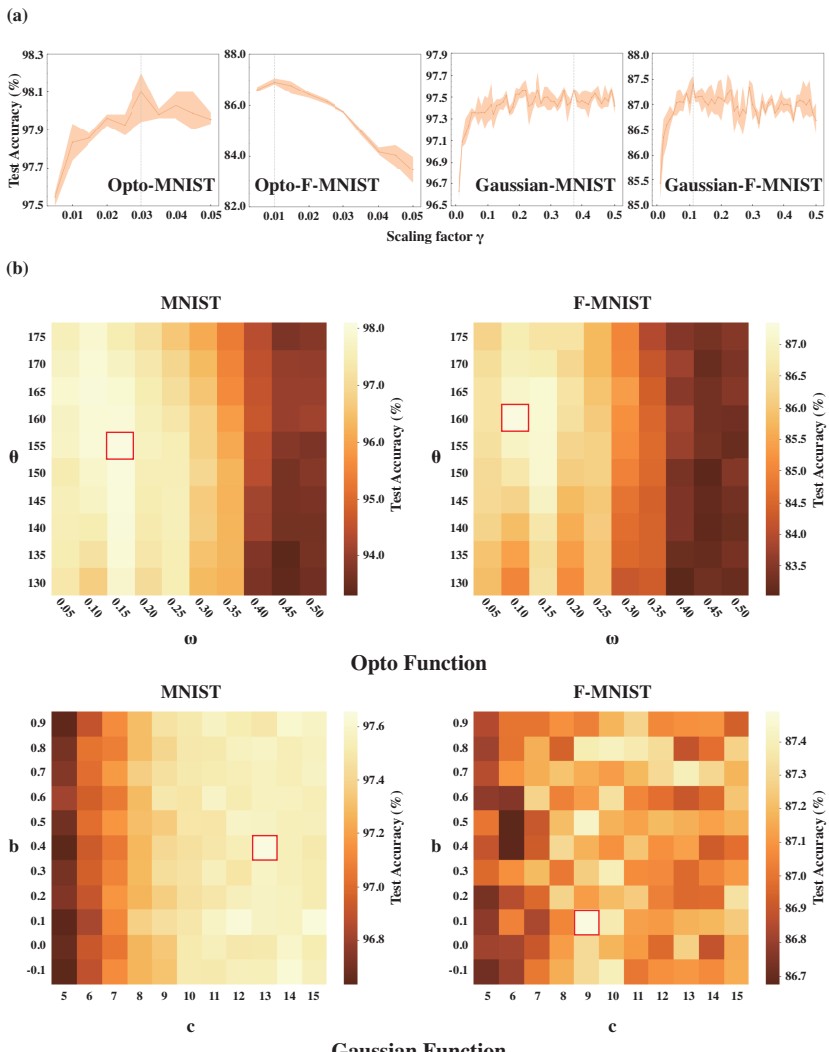

Figure A.4: **The results of optimizing the backward processes of "Opto"-based and Gaussian-based aDFA-SNNs frameworks.** **(a)** The results of adjusting scale factor $\gamma$ in the initialization process of $B$ of both frameworks by employing grid-search. The line charts depict the test accuracies of frameworks as the functions of $\gamma$. The solid line representing the average performance and the shaded region indicating the range between maximum and minimum values. **(b)** The results of adjusting $\theta$, $\gamma$ in "Opto"-based aDFA-SNNs framework and $b$, $c$ in Gaussian-based aDFA-SNNs framework. The color scheme corresponds to the average test accuracy on MNIST and F-MNIST tasks of frameworks. The red box denotes the best performance setting. The "Opto" function based framework can achieve 98.10% and 87.34% test accuracies on MNIST and F-MNIST datasets, the corresponding parameter-combinations are $\omega = 0.15$, $\theta = 155$, as well as $\omega = 0.1$, $\theta = 160$, respectively. The best performances of Gaussian function based framework on MNIST and F-MNIST datasets are 97.66% and 87.49%, the corresponding parameter-combinations are $b = 0.4$, $c = 13$, and $b = 0.1$, $c = 9$, respectively.

alternative to gradients of SNNs due to its characteristic bell curve shape[14, 48], expressed as:

$$g(x) = ae^{\frac{-(x-b)^2}{2c^2}}, \tag{A.11}$$

where $a$ is used to control the height of the function, which is initialized to 1; $b$ is the center of the peak on the x-axis and is initialized to 0.4, which equals to the threshold value of the LIF neurons that we used; $c$ represents the width of "bell", namely the Gaussian RMS width, which is initialized

to 10 because good performance can be obtained at this order of magnitude (the analysis for this part is included in Appendix J).

The other one is called "Opto" function, an optically friendly equation used in the initial aDFA study [26], is shown as:

$$g(x) = cos^2 \left( \omega x + \theta \right),  \qquad (A.12)$$

where $\omega$ is angular frequency, which controls the horizontal extension degree on x-axis and is initialized to 0.1 (the order of magnitude analysis of this parameter is shown in Appendix J); the $\theta$ is phase, defines the position of function, which is initialized to 150 in our cases.

In the adjusting process, first, under the aforementioned initialized settings, we conduct the grid-search on the scale factor $\gamma$ of $B$ of both frameworks respectively (note that in the PRFS experiments, since each backward PRFS has a different shape, for consistency, $\gamma$ is fixed to a constant value of 0.0338). The test accuracy of frameworks on MNIST and F-MNIST datasets are utilized as metrics to identify great-performance points, enabling direct adjustments to be made on $B$. The results of three-layer fully connected SNN model with dimension $784 \times 1000 \times 10$ are shown in Fig.A.4 a. The line charts depict the test accuracy as the functions of $\gamma$, with the solid line representing the average performance and the shaded region indicating the range between maximum and minimum values. As can be seen, great-performance points of $\gamma$ exhibit characteristics of task-specific and framework-specific. That is, the values of $\gamma$ for achieving good accuracy are different when the tasks and the backward function of frameworks are different. In our cases, for "Opto" based framework, the great-performance points of $\gamma$ on both MNIST and F-MNIST datasets can be directly and clearly obtained, with values of 0.03 and 0.01 respectively. While for Gaussian function based framework, the performances are significantly enhanced when $\gamma$ achieving certain ranges. Despite in these ranges, the performances on both MNIST and F-MNIST datasets fluctuate with varying $\gamma$, we still can identify several great-performance points. We chose $\gamma$ values of 0.37 and 0.11 for MNIST and F-MNIST datasets, respectively. These points are chosen as they demonstrate high average accuracy and low data dispersion, which means great and stable performances of the framework.

Then, based on the selected $\gamma$ of $B$, we directly adjust the parameters of backward function $g$ in above frameworks to further improve performances of them. Since the "Opto" and Gaussian functions are only primarily influenced by two key parameters, namely $\omega$, $\theta$ as well as $b$, $c$, respectively. To obtain great parameter-combinations of these functions, we generate heatmaps by utilizing the grid-search approach. The results are shown in Fig.A.4 b, where the color scheme corresponds to the average test accuracy achieved on the given tasks. It is worth noting that the lighter shade of color indicates the higher accuracy. For the "Opto" function based framework, the best performances on MNIST and F-MNIST datasets that we can achieve are 98.10% and 87.34%, the corresponding parameter-combinations are $\omega = 0.15$, $\theta = 155$, as well as $\omega = 0.1$, $\theta = 160$, respectively. For the Gaussian function based framework, the obtained best performances on MNIST and F-MNIST datasets are 97.66% and 87.49%, the corresponding parameter-combinations are $b = 0.4$, $c = 13$, and $b = 0.1$, $c = 9$, respectively.

## J  Preliminary analysis of parameters of "Opto" and Gaussian based frameworks

The horizontal extension degree of the function is an important parameter, which seriously affects the scope and shape of the function in the direction of the x-axis. In our cases, we used the width of the Gaussian function and the width of the 'Opto' function in one period to represent it. For Gaussian function, the $c$ in Eq.A.11 is proportional to the width of function, and for 'Opto' function, the $\omega$ in Eq.A.12 is inversely proportional to the width of the function in one period. In these experiments, we used a fully connected SNN model with dimension of $784 \times 1000 \times 10$. We tested 5 different orders of magnitude of $c$ and $\omega$ from $10^{-2}$ to $10^2$ and checked corresponding test accuracy of these settings on MNIST and F-MNIST tasks, to get a preliminary range of settings for widths that can achieve good performance for subsequent and detailed analysis. The upper two schematics of Fig.A.5 show the results of test accuracy. The blue line and red line represent performance on MNIST task and F-MNIST task, respectively. For Gaussian function, the best performances are achieved when $c$ in the order of $10^1$, with average test accuracies of 97.2% on MNIST and 86.5% on F-MNIST. For 'Opto' function, the best performances are achieved when $\omega$ in the order of $10^{-1}$, with average test accuracy of 97.84% on MNIST and 85.3% on F-MNIST. In order to explain these performances, we

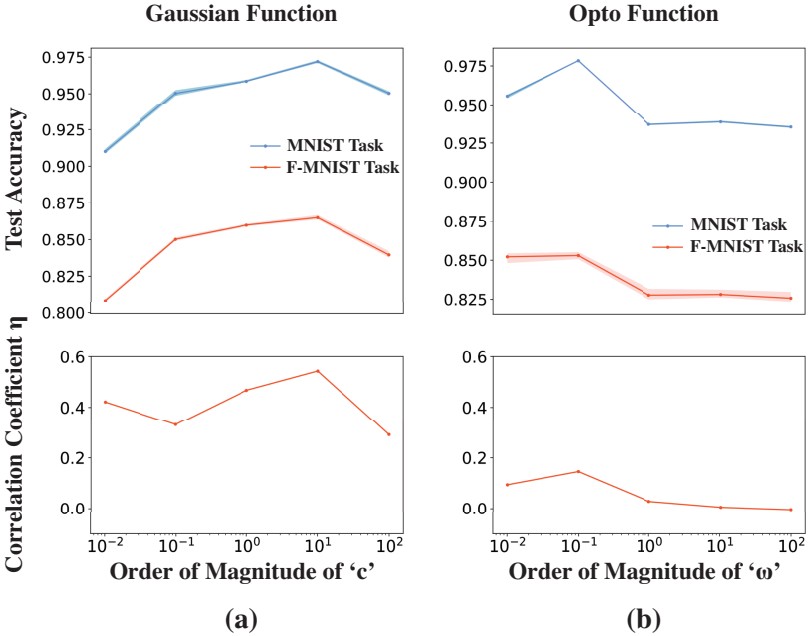

Figure A.5: **The results of analysis of the horizontal extension degree of backward functions** $g$**.** The upper two schematics show the results of the effect of horizontal extension degree of backward functions on performance of aDFA-trained SNNs. The blue line and red line indicate average performance on MNIST task and Fashion-MNIST task respectively, and shaded area indicates maximum-minimum region. The lower two plots represent the corresponding correlation coefficients for the order of magnitude of the horizontal extension degree of the backward function $g$. **(a)** The test accuracy and correlation coefficient as functions of order of magnitude of $c$ in Gaussian function. **(b)** The test accuracy and correlation coefficient as functions of order of magnitude of $\omega$ in 'Opto' function.

also calculated the corresponding correlation coefficients between $f'$ and used $g$, which are Gaussian function and 'Opto' function. The results are shown in the two bottom graphs of Fig.A.5. For both Gaussian function and 'Opto' function, the trends of correlation coefficient $\eta$ with respect to the order of magnitude of parameters are almost identical to the trends of test accuracy, i.e., relatively better performances are obtained at settings with relatively high $\eta$.

However, there are exceptions, for example, the point in the Fig.A.5a, where the magnitude of $c$ is equal to $10^{-1}$, has a relatively low correlation coefficient $\eta$ but relatively high performance compared to other points. As well as for the best performance on MNIST, 'Opto' function's is better than that of Gaussian function, but the corresponding correlation coefficient is lower than Gaussian function's. For these cases, we think it is due to the different input distributions of $x$ of the backward functions $g(x)$. We computed correlation coefficient $\eta$ on the integration interval $[-100, 100]$, but different datasets, different backward functions $g$ and different parameter settings all lead to different input distributions range of $g$ and thus different effective working intervals of $g$, and the parts of the function outside the effective working intervals do not contribute to the training as well as the performance, so the exact calculation of correlation coefficient $\eta$ should be task, function and setup specified.

## K The comparison with the performance of existing studies.

Table.A.1 shows the results of our schemes and existing studies of full-connected SNNs. We also compare them from the perspective of neuromorphic hardware feasibility. We define neuromorphic hardware feasibility in terms of the difficulty of a fully physical implementation of the training algorithm. The "No" implies that full physical implementation is impossible; the " Low" implies the existence of a layer-by-layer error propagation mechanism that is difficult to implement physically,

"Medium" implies BP-free but still requires the design and compute the accurate gradient, "High" stands for algorithms where BP, gradient are both free. Note that in the initial aDFA study[26], although the "Opto" function is utilized to train SNNs on MNIST task, there are no systematic investigation and optimization of the aDFA-SNNs framework. It can be seen that the ANN-to-SNN [13] and BP-Surrogate Gradient [49, 50] methods can achieve highest accuracy, but their physical implementation is challenging. In the DFA based approaches[37, 39, 51], while the relaxed error transmission mechanism can enhance physical implementation feasibility, from the perspective of neuromorphic computing, approximating dynamics of SNN neurons during designing process of the backward function still poses difficulties in their physical implementation. In contrast, our schemes can achieve high feasibility for implementation of neuromorphic hardware lie in their ability of utilizing relaxed nonlinearities rather than complicated design processes. By employing the simplistic, straightforward, and hardware-friendly optimization technique that directly adjust the parameters in the backward process, our frameworks can obtain competitive performances.

| Dataset | Method | Architecture | Neuromorphic Hardware Feasibility | Accuracy |
|---|---|---|---|---|
| **MNIST** | ANN-to-SNN[13] | 784-1200-1200-10 | No | 98.68% |
| | BP-Surrogate Gradient[49] | 784-500-500-10 | Low | 98.70% |
| | BP-Surrogate Gradient[50] | 784-800-10 | Low | 97.55% |
| | BP-STDP[52] | 784-500-150-10 | Low | 97.20% |
| | eRBP(DFA)[51] | 784-500-500-10 | Medium | 97.64% |
| | SNN-BA(DFA)[37] | 784-630-370-10 | Medium | 97.05% |
| | DeepTempo(DFA)[39] | 784-500-500-10 | High | 95.70% |
| | aDFA(Opto)[26] | 784-1000-10 | High | 98.05% |
| | **aDFA-Opto(Ours)** | | | **98.10%** |
| | **aDFA-Gaussian(Ours)** | **784-1000-10** | **High** | **97.66%** |
| | **aDFA-GA-PRFS(Ours)** | | | **97.91%** |
| **F-MNIST** | EM-STDP [53] | 784-500-500-10 | Medium | 86.10% |
| | Global Feedback + STDP[54] | 784-500-500-500-500-500-10 | Medium | 89.05% |
| | sym-STDP[55] | 84-6400-10 | High | 85.31% |
| | **aDFA-Opto(Ours)** | | | **87.34%** |
| | **aDFA-Gaussian(Ours)** | **784-1000-10** | **High** | **87.46%** |
| | **aDFA-GA-PRFS(Ours)** | | | **87.20%** |

Table A.1: **The performance comparisons of proposed aDFA-SNNs frameworks with existing methods on MNIST and Fashion-MNIST tasks.** BP: backpropagation. STDP: spike-timing-dependent plasticity. DFA: direct feedback alignment. aDFA: augmented direct feedback alignment. GA: genetic algorithm. F-MNIST: Fashion-MNIST task. PRFS: positive random Fourier series.

