# OpenReview forum: "Training Spiking Neural Networks via Augmented Direct Feedback Alignment"
_NeurIPS.cc/2024/Workshop/MLNCP — MLNCP Poster_

### Official Review · Reviewer_7spB · 2024-09-24
**Applying aDFA to spiking neural networks**

**Rating:** 7
**Confidence:** 3

**Review:**

- The paper applies augmented direct feedback alignment (aDFA) [26] to the setting of spiking neural networks SNN.
aDFA is an incremental but useful extension of feedback alignment, in which during the backwards pass the derivative of the activation function $f'$ is replaced by an arbitrary function $g$. This is useful in the context of SNNs were $f'$ is either 0 or undefined due to the discrete nature of f. The authors explore a family of functions g and find that aDFA outperforms backprop (backprop with a differentiable surrogate function in place of f').

- I think it is hard to conclude much from the experiments, as the network only has a single hidden layer. Sometimes (often) bio-inspired alternatives to BP perform well on small-scale experiments but fail in deeper models or CNNs (this is the case for feedback alignment).

** Minor comments: **

- Is aDFA really biologically plausible or is it just less implausible than BP. aDFA still relies on transmitting signed error signals backwards, which seems problematic with respect to Dale's principle.

- This sentence does not make sense: "during the process of backward propagating errors, the networks need to fully record carefully orchestrated adjustments of all synaptic weights during forward propagation". Perhaps the last three words should be cut out.

- It is incorrect to call feedback alignment one of the earliest backprop-free algorithms. Feedback alignment is decades younger than e.g., Wake-sleep, recirculation, the perceptron learning rule, contrastive divergence and various Hebbian/anti-Hebbian learning rules (and probably many other BP-free methods).

---

### Official Review · Reviewer_NJVS · 2024-10-04
**The authors evaluated the applicability of augmented direct feedback alignment to SNNs with focus on hardware compatibility**

**Rating:** 7
**Confidence:** 3

**Review:**

Overall, the approach sounds interesting, however, I was wondering, what makes aDFA outperform backprop, while DFA fails to do so. In general, the manuscript was nice, however, I would suggest two additions:

1. Could you evaluate your method on deeper networks? Mostly, shallow networks are easy to train and difficulties only show in deeper nets.
2. It would be nice, if you validate your approach also on typical SNN benschmarks, such as for example the spiking heidelberg digits.

---

### Decision · Program_Chairs · 2024-10-10

Accept (Poster)